# CRISPR/Cas9-Mediated Mutagenesis of Sex-Specific *Doublesex* Splicing Variants Leads to Sterility in *Spodoptera frugiperda*, a Global Invasive Pest

**DOI:** 10.3390/cells11223557

**Published:** 2022-11-10

**Authors:** Junwen Gu, Jingyi Wang, Honglun Bi, Xuehai Li, Austin Merchant, Porui Zhang, Qi Zhang, Xuguo Zhou

**Affiliations:** 1College of Plant Protection, Shenyang Agricultural University, Shenyang 110866, China; 2State Key Laboratory of Cotton Biology, School of Life Sciences, College of Agriculture, Henan University, Kaifeng 475004, China; 3Department of Entomology, University of Kentucky, Lexington, KY 40546, USA

**Keywords:** *Spodoptera frugiperda*, invasive species, *doublesex*, sex determination, CRISPR/Cas9 system, SIT

## Abstract

*Spodoptera frugiperda* (J. E. Smith), an emerging invasive pest worldwide, has posed a serious agricultural threat to the newly invaded areas. Although somatic sex differentiation is fundamentally conserved among insects, the sex determination cascade in *S. frugiperda* is largely unknown. In this study, we cloned and functionally characterized *Doublesex* (*dsx*), a “molecular switch” modulating sexual dimorphism in *S. frugiperda* using male- and female-specific isoforms. Given that Lepidoptera is recalcitrant to RNAi, CRISPR/Cas9-mediated mutagenesis was employed to construct *S. frugiperda* mutants. Specifically, we designed target sites on exons 2, 4, and 5 to eliminate the common, female-specific, and male-specific regions of *S. frugiperda dsx* (*Sfdsx*), respectively. As expected, abnormal development of both the external and internal genitalia was observed during the pupal and adult stages. Interestingly, knocking out sex-specific *dsx* variants in *S. frugiperda* led to significantly reduced fecundity and fertility in adults of corresponding sex. Our combined results not only confirm the conserved function of *dsx* in *S. frugiperda* sex differentiation but also provide empirical evidence for *dsx* as a potential target for the Sterile Insect Technique (SIT) to combat this globally invasive pest in a sustainable and environmentally friendly way.

## 1. Introduction

*Spodoptera frugiperda* (J. E. Smith) (Lepidoptera: Noctuidae), commonly known as fall armyworm (FAW), is an important agricultural pest native to the United States that has spread and become a serious threat to a variety of other countries [1,2]. The species has been divided into a corn strain (CS) and rice strain (RS) according to host preference and genetic difference [3,4]. In China, *S. frugiperda* was first reported in the Yangtze River Valley, bordering Burma, in 2019 [5]. This invasive pest can cause significant losses when infesting crops such as sorghum, cotton, and rice [6,7]. Control of *S. frugiperda* becomes imperative to preventing dramatic loss of crops, and conventional management of *S. frugiperda* relies on the use of chemical agents [8,9,10,11] or transgenic plants harboring *Bacillus thuringiensis* toxins [12,13]. Considering the occurrence of pest resistance to chemical and biological pesticides [14,15], alternative strategies incorporating genetic approaches, such as the Sterile Insect Technique (SIT), are urgently needed for long-term pest control.

*Doublesex* (*dsx*) acts as a regulatory element that controls sexual dimorphism in insects [16]. *Dsx* was first found to produce male- and female-specific proteins via alternative splicing that regulate sex differentiation in *Drosophila melanogaster* [16]. Later, *dsx* homologues were reported to contribute to sexual dimorphism and polymorphism in a number of different insect species, including honey bees, wasps [17,18], flies, mosquitoes [19,20,21,22,23], beetles [24,25,26], and butterflies and moths [27,28]. Sexually dimorphic traits in insects include body patterning, body size, abdominal genitalia, and sex-specific physiology [29]. Expression of *dsx* is initiated in the early embryonic stage and persists until maturity to regulate the development of sexual dimorphic traits [30,31]. Although previous studies show that splicing isoforms of *dsx^F^* and *dsx^M^* dictate sex determination in multiple insects, i.e., somatic sexual differentiation of *dsx* is fundamentally conserved among insects, the sex determination cascade in *S. frugiperda* is still not fully understood.

Functional studies of *dsx* through genetic approaches have recently been reviewed [32]. Silencing of *dsx* by RNA interference led to female-like genitalia development in *Nasonia vitripennis* males [17] and disrupted olfaction and reproduction in *Aedes aegypti* [33]. Most recently, the CRISPR/Cas9-mediated mutagenesis system was applied to address the functionality of *dsx* in Lepidoptera. Knocking out *Plutella xylostella dsx* resulted in malformation of external genitalia and decreased hatchability of eggs [34]. In *Spodoptera litura*, adult male *dsx* mutants showed smaller testes and an inability to mate with wildtype females [35]. A recent study in *Ostrinia furnacalis* also verified the vital role of *dsx* in sexual dimorphism [36]. CRISPR/Cas9-mediated gene editing of *dsx* reduced the size of reproductive organs and fertility in *Apis melifera* [37]. In addition, *dsx* knockout combined with a gene drive system has rapidly spread the disrupted *dsx* allele to elimination of *Anopheles gambiae* population after 7–11 generations in laboratory settings [38].

Factors involved in the sex determination cascade include primary signals, executors, and transducer master regulators [32]. *Dsx* is a highly conserved downstream gene dictating sex differentiation at the bottom of the sex determination cascade [39]. The primary signal of sexual dimorphism in *Bombyx mori* comes from the female-specific factor, PIWI-interacting RNA (piRNA), located on the W chromosome [40]. In the female sex determination cascade, piRNA is controlled by the primary signal *Fem* to target and cleave the downstream gene *Masculinizer* (*Masc*), which encodes the zinc-finger protein masculinizer (MASC) [32]. *Masc* is transcribed from the Z chromosome and plays an important role in masculinization and dosage compensation and regulates the formation of *dsx^M^* in the male sex determination cascade [41,42]. The sex-specifically expressed genes, *Olfactory Receptor* (*OR*) and *Pheromone Binding Protein* (*PBP*), are associated with sexual dimorphism in *O. furnacalis* [43]. Research on identification of *OR* and *PBP* also confirmed that knockout of *dsx* interferes with expression of sexually dimorphic genes [34,36,44].

SIT is a classical genetic pest control strategy used to suppress pest populations in the field by releasing insects carrying sex-specific lethal genes or sex-specific sterilized genes. In insects, *dsx* determines sex differentiation at the bottom of the pathway [39]. In *B. mori*, the functional importance of *dsx* was verified, and its potential application in SIT was proposed [45]. Novel population genetic control methods, gene drives, targeting female-specific lethal genes have been reported in *A. gambiae*, such as a sex distorter targeting *Fle* and a gene drive targeting *dsx*, which caused progressive decreased ratio of female and eventually collapsed population [38,46,47]. Experiments that address alternative splicing to *dsx* would help to test the feasibility of using *dsx* as a target gene for SIT.

In this study, we hypothesized that alternative splicing of *Sfdsx* regulates sex determination and is involved in fecundity and fertility of *S. frugiperda*. The objective of this study was to investigate the functional role of *dsx* in *S. frugiperda*. Through the CRISPR/Cas9-mediated gene-editing system, exons encoding common female- and male-specific transcript of *dsx* were designed for specific knockouts. Impacts on both morphology and physiology were observed.

## 2. Materials and Methods

### 2.1. Insect Rearing and Sexing

The *S. frugiperda* strain used in this study was collected from Dongyang, Zhejiang Province, China. Larvae were kept in the lab within acrylic boxes and fed with artificial diet. The main components of artificial diet were yeast extract, wheat bran, vitamin C, sucrose, and agar as described previously [36]. The strain was maintained under conditions of 25 °C, 70% relative humidity, and a photoperiod of 16:8 h L:D. After pupation, pupae were sexed based on external abdominal characters and kept separated by sex to prepare for pairing. After eclosion, adults were maintained in a plastic bag provisioned with cotton balls soaked with 10% honey water for reproduction.

### 2.2. Molecular Cloning of Sfdsx

Total RNA of day one third-instar *S. frugiperda* was extracted with Trizol reagent (Invitrogen, Carlsbad, CA, USA), and cDNA was synthesized using a GoScript^TM^ reverse transcription kit (Promega, Madison, WI, USA) according to the manufacturers’ instructions. The *dsx* of *S. frugiperda* was identified through blast against the amino acid sequences of *D. melanogaster* (GenBank accession number NP_001287220.1) and *B. mori* (GenBank accession number NP_001036871.1) against *S. frugiperda* in NCBI. Primers were designed using the Primer 3 website (https://primer3.ut.ee, accessed on 1 May 2000) to amplify exons 2 to 5 flanking the coding regions of female- and male-specific regions of *Sfdsx* (Appendix A).

### 2.3. Synthesis of Sfdsx sgRNAs In Vitro

Specific target sites on exons harboring alternative splicing of *dsx* were selected, specifically, sgRNA targeting exon 2 was designed to target the common region, exon 4 for the female-specific region, and exon 5 for the male-specific region. All the gRNAs were designed using an online tool, CRISPRdirect (http://crispr.dbcls.jp/, accessed on 20 November 2014) [48]. We designed gRNAs targeting *Sfdsx^C^*, *Sfdsx^F^*, and *Sfdsx^M^* sites following the rule of 5′-GG-(N)18-NGG-3′ on *dsx*. The *Sfdsx^C^*, *Sfdsx^F^*, and *Sfdsx^M^* gRNA sequences were designed with a length of 20 bp and aligned with *S. frugiperda* genome sequence (ASM1297921v2) to determine their specificity. The sgRNAs were sub-cloned and ligated to the pJET1.2 vector (ThermoFisher Scientific, Waltham, MA, USA). sgRNAs were synthesized using MEGAScript T7 (Ambion, Austin, TX, USA) in vitro following the manufacturer’s instructions. The TrueCut^TM^ Cas9 Protein (Invitrogen, Carlsbad, CA, USA) was purchased commercially and stored at −80 °C for experimental use.

### 2.4. Embryo Microinjection

To collect eggs for microinjection, five pairs of *S. frugiperda* adults were sexed and paired in a transparent plastic bag. Collected eggs were injected with 300 ng/μL Cas9 protein mixed with 300 ng/μL sgRNA within 1 h of oviposition under microscope (Olympus ZSX16, Tokyo, Japan). After injection, the eggs were incubated at 25 °C for 4 days until hatching and transferred to containers with artificial diet.

### 2.5. Genomic DNA Extraction and Mutagenesis Analysis

Genomic DNA of the pupal shell was extracted using phenolic chloroform and precipitated with isopropanol sodium acetate by incubation with protease K (ThermoFisher Scientific, Waltham, MA, USA). Primers were designed to amplify the spanning region of the three knockout sites using Hieff Canace Gold High Fidelity DNA Polymerase (Yeasen, Shanghai, China), following these reaction conditions: 98 °C 3 min pre-denaturation; 98 °C 10 s, 55 °C 20 s, 72 °C 30 s, altogether 35 cycles. After 72 °C 30 s of final extension, the product was connected to the pJET1.2 vector (ThermoFisher Scientific, Waltham, MA, USA) and sent to Sangon Biological Company for sequencing. The phenotype of the mutant was photographed using a micro-imaging system (KEYENCE VHX7000, Osaka, Japan).

### 2.6. Phenotypic Impacts of Mutagenesis

The phenotypic impact on morphology of pupae and adults, including external genitalia, testes, and ovaries, was imaged under a stereo microscope (KEYENCE VHX7000, Osaka, Japan). The number of injected eggs used for gene knockout, hatching rate, pupation rate, and sex ratio of molted adults were recorded and summarized in Table 1.

To investigate the impact of *dsx* mutagenesis on adult fecundity and sterility, two-day-old *dsx* mutants were paired with opposite-sex adults. *Sfdsx^C^*, *Sfdsx^F^*, and *Sfdsx^M^* mutants were paired with opposite-sex WT virgin females or males, while WT pairings were used as a control. Each pair was maintained in a single plastic bag supplemented with 10% honey water. Eggs were collected, and their number was recorded each day for ten days post-pairing. Afterward, egg hatching rate was recorded for each pair. Experiments were performed three times with three to five pairs for each treatment.

### 2.7. Real Time Quantitative PCR (RT-qPCR)

After sex-specific mutagenesis, total RNA was extracted from the whole body of adults using Trizol reagent (Invitrogen, Carlsbad, CA, USA), including wildtype females and males, *Sfdsx^C^* males and females, *Sfdsx^F^* females, and *Sfdsx^M^* males. cDNAs was synthesized using 1 μg total RNA as template using a GoScript^TM^ reverse transcription kit (Promega, Madison, WI, USA) according to the manufacturers’ instructions. To investigate the effects of *Sfdsx* mutagenesis on the *OR* and *PBP* genes, RT-qPCR was performed to examine the relative expression of *OR1*, *PBP1*, and *PBP2* after *Sfdsx* was mutated. *ß-actin* was the internal reference [49]. The reaction conditions were pre-denaturation at 98 °C for 3 min; 98 °C 10 s, 55 °C 20 s, 72 °C 30 s, 40 cycles in total; 72 °C for 5 min using a Hieff canal gold high-fidelity DNA polymerase kit (Yeasen, Shanghai, China) for gene amplification.

To investigate the temporal expression of *Sfdsx* among different developmental stages, total RNA from *S. frugiperda* eggs, day one first to sixth instar larvae, wandering stage larvae, pupae, female adults, and male adults was extracted using Trizol reagent (Invitrogen, Carlsbad, CA, USA). A total of 5 μg RNA was used as a template, and cDNA was synthesized with a GoScript^TM^ reverse transcription kit (Promega, Madison, WI, USA). Quantitative real time PCR was performed to examine the temporal expression of *Sfdsx,* and *ß-actin* was the internal reference. Relative gene expression was calculated following the 2^−ΔΔCT^ method [50]. Statistical analysis was performed using IBM SPSS statistics 22 software using a two-tailed *t*-test. Column and error bars stand for mean ± SEM in all cases. *p* < 0.05 was considered a significant difference.

## 3. Results

### 3.1. Identification and Cloning of Sfdsx

The genomic sequence length of *Sfdsx* was 196,795 bp, consisting of six exons with two sex-specific splicing patterns. The length of *Sfdsx* female-specific ORFs was 780 bp, encoding 259 amino acids. The length of male-specific ORFs was 801 bp, encoding 266 amino acids. The predicted alternative splicing patterns are shown in Appendix A. We used the NCBI BLAST program to identify *Sfdsx* against homologous amino acid sequences in *D. melanogaster* (GenBank accession number NP_001287220.1) and *B. mori* (GenBank Accession number NP_001036871.1).

### 3.2. CRISPR/Cas9-Mediated Sfdsx Mutagenesis

We selected one target site for mutagenesis within the common, female- and male-specific regions on exon 2, 4, and 5, respectively. Exon 2 was predicted to encode a common region of male and female transcripts, exon 4 a female-specific isoform, and exon 5 a male-specific isoform (Figure 1A). We injected 815, 702, and 1325 embryos within 1.5 h post-oviposition with Cas9 protein mixed with pre-synthesized *Sfdsx^C^*, *Sfdsx^F^*, and *Sfdsx^M^* sgRNAs, respectively. The hatching and mutant ratios for each treatment are recorded in Table 1. We obtained pupal mutant ratios of 38.5%, 35.4%, and 35.1% for *Sfdsx^C^*, *Sfdsx^F^*, and *Sfdsx^M^* treatments, respectively (Table 1). After eclosion, we found a significantly male-biased sex ratio in *Sfdsx^C^* mutants in comparison to wildtype adults. In *Sfdsx^F^* and *Sfdsx*^M^ mutants, morphological changes were observed exclusively in the corresponding sex of sex-specific knockouts (Table 1). To identify the mutated alleles, we extracted genomic DNA from pupa shell, and the results of genomic sequencing showed successful deletion of nucleotides within the target sites in *Sfdsx^C^*, *Sfdsx^F^*, and *Sfdsx^M^* isoforms (Figure 1C). To investigate the temporal expression of *Sfdsx* in different developmental stages, relative expression levels of *Sfdsx* in eggs, first to sixth instar larvae, wandering stage larvae, pupae, and adults were examined. Results showed that the relative expression level of *Sfdsx* was significantly increased at the pupal stage but showed no significant change among the other stages (Figure 1D).

### 3.3. Phenotypic Impacts of Sfdsx Mutagenesis

#### 3.3.1. External Genitalia

Phenotypic changes brought by gene knockout were recorded. We distinguished the sex of pupae based on external morphological characteristics, specifically, male pupae exhibit two noticeable points in the middle of the ninth abdominal segment, while female pupa exhibit two “^”-shaped lines across the eighth abdominal segment (Figure 2). In *Sfdsx^C^* injections, all female and male pupal mutants exhibited abnormal external genitalia on the abdomen (Figure 2). All male pupae in the *Sfdsx^F^* knockout group and all female pupae in the *Sfdsx^M^* knockout group were externally similar to the wildtype (Figure 2). The corresponding phenotypic changes of *Sfdsx^F^* and *Sfdsx^M^* knockout groups were similar with the phenotypic changes found in *Sfdsx^C^* mutants. In almost all *Sfdsx^F^* mutants, the two “^”-shaped lines became absent with other irregular structural changes located across the eighth and ninth segments, while *Sfdsx^F^*-injected males were identical to the wildtype. For *Sfdsx^M^* mutants, abnormally shaped and numbered masses were found on both sides of the gonopore on the eighth to ninth abdominal segment in males, but females were not affected (Figure 2). The figure representing the gonopore in males is also obfuscated after *Sfdsx^M^* mutagenesis (Figure 2).

Female and male pupae and adults of *S. frugiperda* are morphologically different and can be distinguished based on the structure of the gonads on the eighth segment of the abdomen (Figure 3). In *S. frugiperda*, the external genitalia of wildtype male adults consist of a pair of harpago, an aedeagus, and an uncus, while the ventral plate and genital papilla are female-specific (Figure 3). In the *Sfdsx^C^* mutants, there were both male and female mutants, and the mutant phenotypes were similar to those of *Sfdsx^F^* and *Sfdsx^M^* mutants. Male *Sfdsx^C^* mutants exhibited a deformed aedeagus, and female *Sfdsx^C^* mutants showed disrupted genital papillae and a deformed ventral plate (Figure 3). *Sfdsx^F^* mutants showed a severely disrupted reproductive ovipositor. In *Sfdsx^M^* mutants, the harpago was severely deformed or developed incompletely, or even completely disappeared, and nodular growth appeared around the external genitalia of some mutants (Figure 3). We did not observe any female-specific external structures in *Sfdsx^M^* mutants or male-specific external structures in *Sfdsx^F^* mutants. This observation suggests that the sex-specific structural development of external genitalia is regulated by *Sfdsx*.

#### 3.3.2. Internal Genitalia

*Sfdsx* mutants with phenotypic changes in the adult stage were dissected to observe internal genital structures. Three days after eclosion, the ovaries of females and testes of males from wildtype and *Sfdsx* adult mutant groups were dissected. As shown in Figure 4, wildtype females have a pair of ovaries, each of which is supplemented with four symmetrical lateral oviducts. Wildtype males have a single fused, rounded testis (Figure 4).

In *Sfdsx^C^* mutants, both females and males exhibited severe malformation of the ovaries and testes (Figure 4). In *Sfdsx^C^* female mutants, the oviduct was severely deformed, possessing four asymmetrical oviducts (Figure 4). The number of premature eggs was highly decreased in the oviducts of *Sfdsx^C^* female mutants. The testes of *Sfdsx^C^* male mutants appeared as irregular spheres with smooth surfaces, having less adhesion to the trachea than those wildtype males, suggesting incomplete development of testes (Figure 4).

In *Sfdsx^F^* mutants, the ovaries of females exhibited obvious oviduct malformation. Two oviducts harboring from one side of the ovary were dramatically shortened in comparison to the other oviducts and those from wildtype ovaries (Figure 4). The *Sfdsx^F^* male mutants showed normal testes compared with wildtype males (Figure 4).

In *Sfdsx^M^* mutants, females showed a normal phenotype. However, the testes in male mutants were smaller in size and exhibited a disrupted structure, indicating failed fusion of the testes during development. Among these mutant phenotypes, gonadal abnormalities were found in male *Sfdsx^M^* mutants and *Sfdsx^C^* mutants of both sexes.

### 3.4. Physiological Impacts

Sex-specific genitalia are important for successful copulation between female and male adults. To investigate if sex-specific *Sfdsx* mutagenesis induces adult sterility, fecundity and hatching rate were recorded continuously for ten days after pairing individuals from different combinations of treatment groups. In wildtype pairings, females lay an average of 1221 ± 84 eggs across the ten days (Figure 5). Pairing *Sfdsx^F^* males and *Sfdsx^M^* females showed no significant differences in fecundity compared to wildtype pairs, demonstrating that the sex-specific exon mutant did not affect the fertility of *Sfdsx^F^* males or *Sfdsx^M^* females. However, fecundity by *Sfdsx^C^* male mutants paired with wildtype female was decreased by approximately 72% to 374 ± 120 eggs on average (Figure 5A), and the hatching rate of these eggs was almost zero (Figure 5B). In *Sfdsx^F^* female mutants, fecundity was decreased by approximately 80% to 249 ± 101 eggs on average (Figure 5A), and the eggs were unable to hatch normally (Figure 5B). *Sfdsx^M^* males paired with wildtype female showed an approximately 91% decrease in fecundity, producing an average of 109 ± 29 eggs (Figure 5A), none of which hatched (Figure 5B).

### 3.5. Pleiotropic Impacts

OR1, PBP1, and PBP2 are pheromone receptors and are involved in mating and reproductive behavior in adults. The expression of the *OR1* was significantly downregulated in *Sfdsx^C^* mutant males and upregulated in *Sfdsx^F^* mutant females (Figure 6A). *PBP1* and *PBP2* were also significantly downregulated in *Sfdsx^C^* mutant males, but expression of *PBP1* and *PBP2* showed no significant difference among *Sfdsx^M^* mutant males (Figure 6A, B). All three genes exhibited decreased transcript level in *Sfdsx^C^* male mutants relative to the wildtype control group, and the transcript levels of *OR1* and *PBP1* were dramatically upregulated in *Sfdsx^F^* female mutants compared to wildtype females (Figure 6). The relative transcript level of *PBP1* was also significantly downregulated in *Sfdsx^C^* female and male mutants, indicating its vital role for both male and female pheromone recognition (Figure 6B).

### 3.6. Sex-Specific Expression of Sfdsx

Primers amplifying the *Sfdsx* female-specific isoform and male-specific isoform show alternative splicing to *Sfdsx* (Appendix A). In wildtype adults, the female-specific isoform is longer (481 bp) than the male-specific band (232 bp), suggesting alternative splicing of the *Sfdsx* (Appendix A). Although the bands of wildtype females and males appear in one lane, transcript levels of *Sfdsx^F^* female-specific and *Sfdsx^M^* male-specific isoforms decreased to a dramatically low level, and the faint bands with a fairly small size suggest successful gene knockout in both *Sfdsx^F^* female and *Sfdsx^M^* male mutants (Appendix A).

## 4. Discussion

*Dsx* has been identified as a downstream gene of the sex determination pathway, playing an important role in the development of sexually dimorphic traits in a variety of insects [51,52]. The functional role of *dsx* has been studied in *Hyphantria cunea* [44], *P. xylostella* [34], *A. gambiae* [38], *Bicyclus anynana* [53], *B. mori* [28], *Agrotis ipsilon* [54], *A. mellifera* [37,55], and *S. litura* [35]. Given that lepidopterans are recalcitrant to RNAi, CRISPR/Cas9-mediated mutagenesis was employed to decipher the functionality of *dsx* in *S. frugiperda. Sfdsx* consists of six exons, of which exons 3 and 4 are female-specific, exon 5 is male-specific, and exon 6 does not participate in transcription (Figure 1A). The nucleic acid sequences of *dsx* from *D. melanogaster* (GenBank accession number NP_001287220.1) and *B. mori* (GenBank Accession number NP_001036871.1) were used as query to identify the putative coding sequence of *Sfdsx*. We cloned female- and male-specific regions of *Sfdsx* and identified one male-specific and three female-specific *dsx* transcripts in *S. frugiperda* (Appendix A). The alternative splicing is similar to that in *S. litura*, of which exons 1 and 2 encode the common region, exons 3 and 4 encode the female-specific region, and exon 5 encodes the male-specific region [35]. Similar splicing patterns were also found in *P. xylostella*, in which one male-specific and three female-specific transcripts were identified [34]. Temporal expression of *Sfdsx* was examined in *S. frugiperda*, and results showed that *Sfdsx* was highly expressed in the pupal stage relative to the other developmental stages (Figure 1D).

Successful gene knockout through the CRISPR/Cas9 gene-editing system was achieved by targeting the common, male-, and female-specific regions of *Sfdsx* (Figure 1B,C). Sex-specific mutagenesis disrupted the development of genitalia in the pupal and adult stages (Figure 2 and Figure 4). In *Sfdsx^C^* mutants, male pupae showed abnormal phenotypes at the eighth abdominal segment, which is consistent with what was observed in *Sfdsx^M^* mutants (Figure 3). The same phenomenon was found in *O. furnacalis*, in which *dsx^C^* mutants displayed an ectopic ventral plate and malformed genital structure [36]. Knockout of *dsx^M^* and *dsx^F^* results in an abnormal abdominal structure at the pupal stage (Figure 2) and malformed genitalia at the adult stage (Figure 3). These phenotypic observations of external genitalia are consistent with the results of *dsx* manipulations in *A. gambiae*, in which disruption of exon 5 led to an intersex phenotype and sterility in females but did not affect development or fertility in males [38]. A male biased sex ratio among *Sfdsx^C^* mutants was observed in comparison to the wildtype adults; this bias might be caused by the lethal effect of *Sfdsx^C^* sgRNAs against females during the embryonic stage [56] (Table 1). In *Sfdsx^C^* and *Sfdsx^F^* female mutants, the oviducts were dramatically shortened or malformed in comparison to those of wildtype females, while in *Sfdsx^M^* mutants, the testes in male mutants were smaller in size and possessed an irregular spherical shape (Figure 4). These observations are consistent with what has been observed in *P. xylostella*, in which smaller testes and disrupted structure of the ovaries were detected in *dsx* mutants [34]. We found that mutagenesis of either the common, female-, or male-specific regions of *dsx* in *S. frugiperda* resulted in reduced fecundity (Figure 5). Similarly, mutagenesis of *dsx* targeting the common region also caused decreased fecundity and fertility of adults in *O. furnacalis* [36]. Reduced fertility after *dsx* mutagenesis was also observed in *P. xylostella* [34]. Our result is in accordance with previous publications showing that *dsx* is associated with sex determination, fecundity, and fertility in insects.

*Dsx* is located downstream of the insect sex determination pathway, plays an important role in morphological development of insect sexual dimorphism, and has a profound impact on insect physiology and behavior [57,58]. As is known in *A. aegypti*, *dsx* not only affects female genital development but also shortens the length of females’ antennae and antennal receptors [33]. In *H. cunea*, *dsx* also affects the development of adult somites [44]. Our result concerning transcript changes of *OR1*, *PBP1*, and *PBP2* in *Sfdsx* mutants showed that all three genes exhibited decreased transcript levels in *Sfdsx^C^* male mutants and dramatically increased levels in *Sfdsx^F^* female mutants compared to the wildtype (Figure 6). Alteration of the expression of pheromone genes may also be a factor affecting mating and oviposition of *S. frugiperda* adults, resulting in the infertility of *S. frugiperda* mutant adults (Figure 5). In *D. melanogaster*, male adults produce courtship songs by vibrating their wings, a behavior that is regulated by *dsx* [59]. At the same time, *dsx* regulates abdominal pigment deposition [51] and *Drop* transcription to affect the normal development of male genital discs [60].

SIT is an environmentally friendly pest management strategy that hinders pest propagation by releasing large numbers of sterile insects into the wild [61,62]. Severe mutations in the external genitalia as a result of *dsx* manipulation have been reported in multiple Lepidopterans [34,36,44]. In our study, knocking out sex-specific *dsx* splicing variants in *S. frugiperda* led to sterility, with the number of eggs produced reduced by 72% in pairings including *Sfdsx^C^* male mutants, 80% in those including *Sfdsx^F^* female mutants, 91% in those including *Sfdsx^M^* male mutants (Figure 5A). Eggs produced by pairings involving *Sfdsx* mutants did not hatch (Figure 5B). Normally, wildtype males display courtship behavior and approach females actively; however, in *Sfdsx^C^* and *Sfdsx^M^* males, the aedeagus structure was shortened, distorted, or even completely absent, which prevented copulation. In *Sfdsx^F^* females, the genital papillae and ventral plate were deformed, also preventing copulation. Eggs laid by *Sfdsx* mutants were dried and ceased development after one to two days, resulting in no hatching (Figure 5). Similarly, a recent study of *O. furnacalis* and *P. xylostella* have shown that eggs laid by *dsx* mutants fail to hatch, suggesting the role of *dsx* as a potential target for genetic control [34,36]. In recent years, complete collapse of laboratory populations of *A. gambiae* through sex-specific sterility has been achieved [46], and complete control of laboratory populations of *A. gambiae* was enabled by targeting *dsx* [38]. We identified loci that cause genital malformations in sex-specific regions of *Sfdsx*. Our study provides a genetic basis for CRISPR gene drive-mediated population suppression and suggests the potential of targeting *Sfdsx* for genetic control of *S. frugiperda*.

## 5. Conclusions

In this study, our combined results support our hypothesis that alternative splicing of *Sfdsx* regulates sex determination and is involved in the fecundity and fertility of *S. frugiperda*. Mutants exhibited substantial external genital abnormalities and distortion of internal genitalia. The fecundity and hatching rate of eggs produced by corresponding mutants was significantly compromised in comparison to wildtype controls. Our results provide empirical evidence of a genetic basis for potentially targeting *dsx* as a part of the Sterile Insect Technique (SIT) to combat this global invasive pest in a sustainable and environmentally friendly way.

## Figures and Tables

**Figure 1 cells-11-03557-f001:**
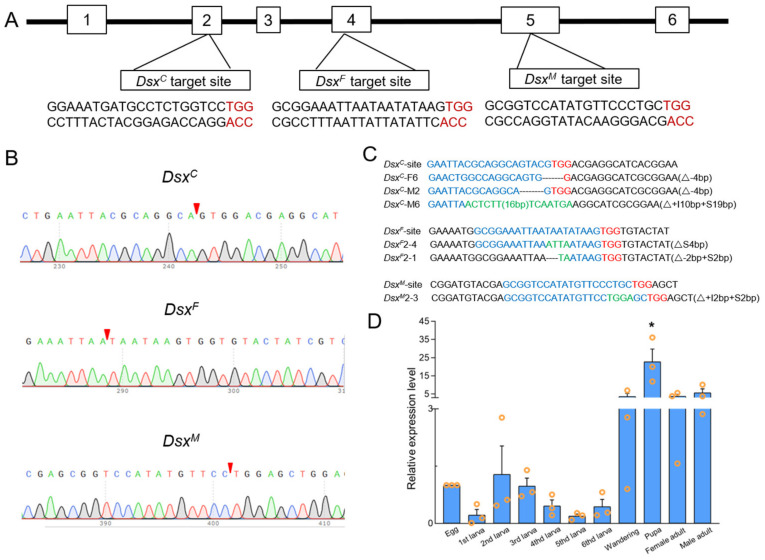
CRISPR/Cas9-mediated mutation within *Sfdsx* target sites and temporal expression of *Sfdsx* gene. (**A**) Target sites on exons 2, 4, and 5 to eliminate the common, female-specific, and male-specific regions of the *Sfdsx* gene. Red represents PAM sequences, and black represents substitutive sequences. (**B**) Sequencing chromatogram of the *Sfdsx^C^*, *Sfdsx^F^*, and *Sfdsx^M^* mutants. The red arrow refers to the position of PCR and sequencing on common, female-specific, and male-specific regions of *Sfdsx* by the CRISPR/Cas9 gene-editing system. (**C**) Common, female-specific, and male-specific mutants of *Sfdsx* detected by sequencing. The dashed lines “–” indicate deleted nucleotides relative to wildtype. The guide RNA target sequences are marked in blue. (**D**) Temporal expression of *Sfdsx* across different developmental stages of *S. frugiperda*. The internal reference is *ß-actin*. The asterisks (*) indicate significant differences (*p* < 0. 05). Data were analyzed using Tukey’s test (one-way ANOVA).

**Figure 2 cells-11-03557-f002:**
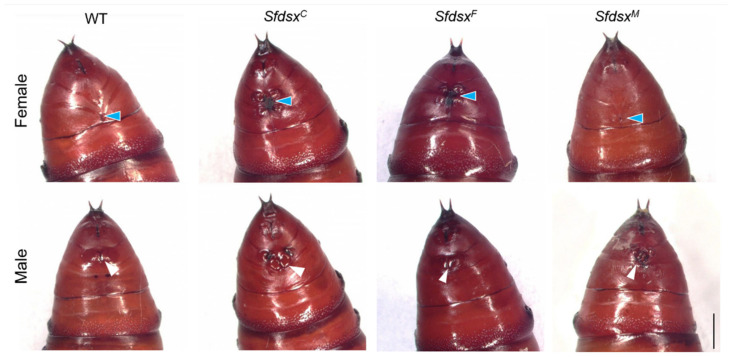
Malformed genital structure after *Sfdsx^C^*, *Sfdsx^F^*, and *Sfdsx^M^* mutagenesis. Photographs of typical wildtype and mutant pupae. Oviposition holes are present on female pupa on the 8th abdominal segment. A longitudinal fissure representing the gonopore is present in the middle of the 9th abdominal segment of the WT male pupa. *Sfdsx^C^*-F, *Sfdsx^C^*-M, *Sfdsx^F^*-F, and *Sfdsx^M^*-M showed abnormal morphology of the gonopore compared with that in WT. Blue arrows depict oviposition holes in females, while white arrows indicate gonopores in males. Scale bars: 1 mm.

**Figure 3 cells-11-03557-f003:**
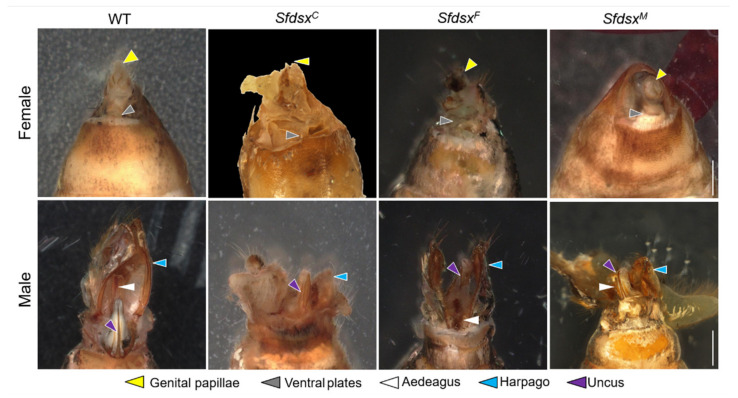
Deformation of external genitalia in *Sfdsx^C^*, *Sfdsx^F^*, and *Sfdsx^M^* mutants at adult stage. The WT male-specific adult external structure consists of a pair of harpago, an aedeagus, and an uncus, whereas the female-specific external genitals include genital papillae and a ventral plate. Compared with WT, the external genitalia of *Sfdsx^C^*, *Sfdsx^F^*, and *Sfdsx^M^* mutants were abnormal. Different color of triangles represents sex-specific external genital structures. Scale bars: 1 mm.

**Figure 4 cells-11-03557-f004:**
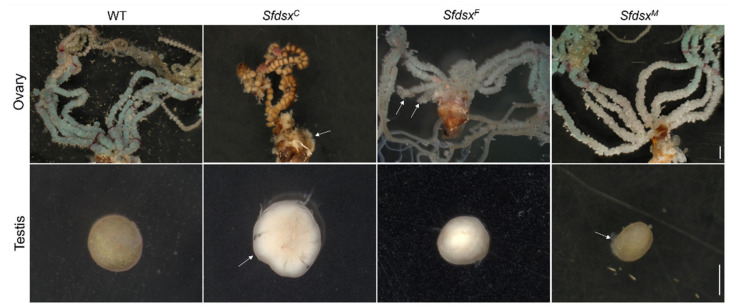
Malformation of ovaries and testes in adult *Sfdsx* mutants. The ovaries and testes of wildtype, *Sfdsx^C^*, *Sfdsx^F^*, and *Sfdsx^F^* adults were dissected on the third day post-eclosion (PAE3). White arrows indicate mutations on ovaries or testes. Scale bars: 1 mm.

**Figure 5 cells-11-03557-f005:**
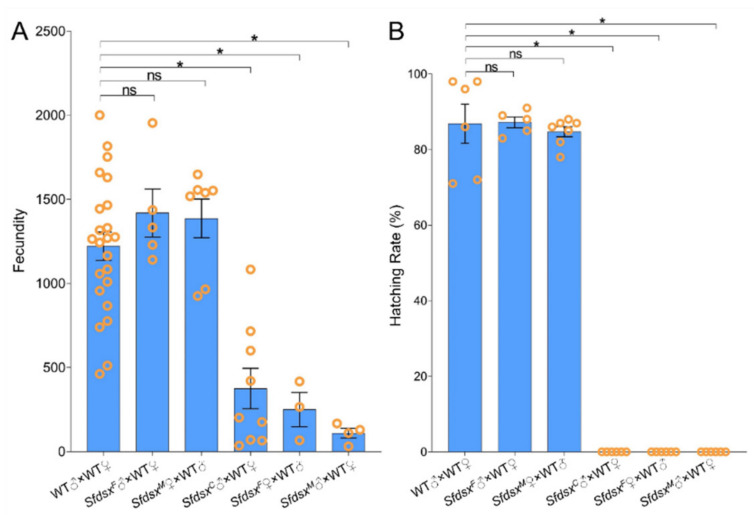
Compromised fecundity and fertility after *Sfdsx* mutagenesis. (**A**) Adult mutants of *Sfdsx* injected with common, female-, and male-specific sgRNAs and Cas9 showed decreased fecundity compared to the wildtype. (**B**) The hatching rates of eggs laid by *Sfdsx^C^* males, *Sfdsx^F^* females, and *Sfdsx^M^* males are shown. The asterisks (*) indicate significant differences (*p* < 0.05) between mutants and wildtypes. “ns” stands for “not significant” (*p* > 0.05). Data were analyzed using a two-tailed *t*-test.

**Figure 6 cells-11-03557-f006:**
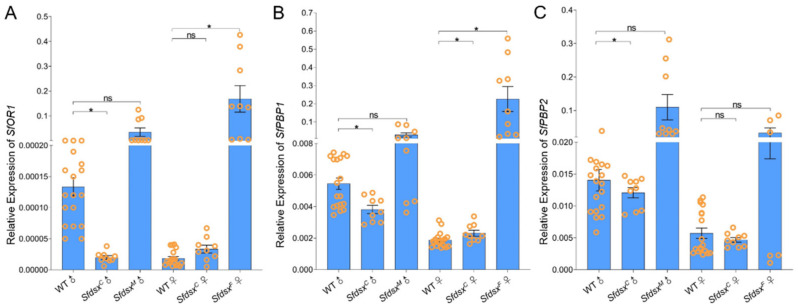
Relative expressions of *OR1*, *PBP1*, and *PBP2* after *Sfdsx^C^*, *Sfdsx^F^*, and *Sfdsx^M^* mutagenesis. The relative transcript levels of *OR1* (**A**), *PBP1* (**B**), and *PBP2* (**C**) were examined in WT male and female, *Sfdsx^C^* male, *Sfdsx^F^* female, and *Sfdsx^M^* males. *ß-actin* was the internal reference. The asterisks (*) indicate significant differences (*p* < 0.05) between mutants and wildtypes. “ns” stands for “not significant” (*p* > 0.05). Data were analyzed using a two-tailed *t*-test.

**Table 1 cells-11-03557-t001:** Mutagenesis of *Sfdsx* induced by CRISPR/Cas9 system.

sgRNA	Injected ^1^	Hatched ^2^	Pupate ^3^	P Mutant ^4^	Adult (F/M) ^5^	A Mutant (F/M) ^6^
*dsx^C^*	815	585 (71.8%)	130 (22.2%)	50 (38.5%)	117 (39/78)	48 (1/47)
*dsx^F^*	702	449 (64.0%)	79 (17.6%)	28 (35.4%)	73 (31/42)	22 (22/0)
*dsx^M^*	1325	461 (34.8%)	37 (6.9%)	13 (35.1%)	36 (17/19)	12 (0/12)
*WT*	776	665 (85.7%)	469 (70.5%)	-	366 (177/189)	-

^1^ Number of injected individuals, ^2^ Number and percent (%) of hatched individuals, ^3^ Number and percent (%) of larvae that entered pupal stage, ^4^ Number and percent (%) of pupal mutants based on morphological change, ^5^ Number and percent (%) of pupae and sex ratio that entered adult stage, ^6^ Number and percent (%) of adult mutants and sex ratio based on morphological change.

## Data Availability

All the data and resources generated for this study are included in the article and the Appendix A.

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
