# Peer review of "CRISPR/Cas9-Mediated Mutagenesis of Sex-Specific Doublesex Splicing Variants Leads to Sterility in Spodoptera frugiperda, a Global Invasive Pest"

_cells, 2022, doi:10.3390/cells11223557_

Round 1
Reviewer 2 Report
This paper described a CRISPR-Cas9 method to target the sex determination gene “dsx” in S. frugiperda. They observed that knocking out sex-specific splicing variants in S. frugiperda led to significantly reduced fecundity and infertility. Their results confirmed the conserved function of dsx in S. frugiperda sex differentiation. Furthermore, their manuscript provides the solid evidence to target dsx as a potential site for Sterile Insect Technique (SIT) to combat this global invasive pest. The organization and writing of this paper are good, and I think it can be published in Cells with a minor revision.
1. It is a little confused that which generation the authors used to evaluate the gRNA efficiency? F0 or F1? How many samples they chose? If F0, how about the efficiency of each gRNA site?
2. For Fig 1B, the authors should indicate which sequence this sequencing chromatogram corresponds to, and in figure legend, they said “The red arrow refers to the position of cleavage”, I think this is not true. Generally speaking, the cas9 will cuts DNA at 3bp upstream of PAM sequence, so I do not know what is the authors mean.
3. For Fig 1D, there lost some information about the labeling of Y-axis.
Author Response
Reviewer 2:
General comments:
This paper described a CRISPR-Cas9 method to target the sex determination gene “dsx” in S. frugiperda. They observed that knocking out sex-specific splicing variants in S. frugiperda led to significantly reduced fecundity and infertility. Their results confirmed the conserved function of dsx in S. frugiperda sex differentiation. Furthermore, their manuscript provides the solid evidence to target dsx as a potential site for Sterile Insect Technique (SIT) to combat this global invasive pest. The organization and writing of this paper are good, and I think it can be published in Cells with a minor revision.
RESPONSE: We are grateful for the enthusiastic comments.
Specific comments:
It is a little confused that which generation the authors used to evaluate the gRNA efficiency? F0 or F1? How many samples they chose? If F0, how about the efficiency of each gRNA site?
RESPONSE: We used F0 generation to evaluate the gRNA efficiency. Pupae with phenotypic changes on the abdomen were pictured, and 5-10 pupae shells were collected to extract genomic DNA to test mutagenesis. The mutant ratio in pupal stage was 38.5, 35.4, and 35.1% for SfdsxC, SfdsxF, and SfdsxM gRNAs, respectively.
For Fig 1B, the authors should indicate which sequence this sequencing chromatogram corresponds to, and in figure legend, they said “The red arrow refers to the position of cleavage”, I think this is not true. Generally speaking, the cas9 will cuts DNA at 3bp upstream of PAM sequence, so I do not know what is the authors mean.
RESPONSE: Following reviewer’s suggestions, we revised the figure legend as follows:
“The red arrow refers to the position of mutation detected by PCR and sequencing on common-, female-specific, and male-specific regions of Sfdsx by CRISPR/Cas9 gene editing system”.
For Fig 1D, there lost some information about the labeling of Y-axis.
RESPONSE: Fig 1D depicted temporal expression of Sfdsx gene across different developmental stages. The missing information concerning the labeling of Y-axis might be caused by the use of “break” function in creating the graph. Given the significant difference of expression levels among the developmental stages, we elected to use this function to reduce the “masking” effect caused by the exceedingly high level of expressions in certain life stage(s), in this case, it was the pupa stage. If reviewer prefers the traditional way, we can eliminate the breaks in Figure 1D.
Reviewer 3 Report
In this study, the authors identified an essential sex determination gene, doublesex (dsx), in a globally important pest Spodoptera frugiperda. The biological functions of dsx has also been explored with CRIPSR/Cas9 genome editing, showing abnormal phenotypes and reduced fertility/fecundity in mutant individuals. Further evidence indicated that these reproduction deficiency was likely caused by down-regulation of pheromone receptors. This research provided potential target for genetic control of S. frugiperda in the future.
The experiments are reasonably designed and data looks basically solid to support authors’ conclusions. The writing of this manuscript is clear in general, but still requires some revisions.
A general issue, the functions of dsx have been reported in many insects including model insects, disease vector mosquitoes and agricultural lepidopteran pests. The results of this study look consistent with those in other insects, thus show less innovation and creativity. It would be more interesting to readers if the authors can discuss more about the differences or potential applications of dsx in S. frugiperda, which will emphasize the importance of this study.
Another question, was there any special phenotype or fitness difference in knockout/wildtype heterozygotes?
Line 17: “Spodopteran” should be corrected as “Spodoptera”.
Line 26-27: “Interestingly, knocking out sex-specific splicing variants in S. frugiperda led to significantly reduced fecundity and infertility, i.e., sterility.” can be revised to “Interestingly, knocking out sex-specific dsx variants in S. frugiperda led to significantly reduced fecundity and fertility in adults of corresponding sex”.
The authors use Sfdsxm, SfdsxF and Sfdsxc in the main context to indicate mutants caused by injecting corresponding gRNAs and Cas9, but ΔSfdsxm, ΔSfdsxF and ΔSfdsxc in figures. Please unify and use the same expression style.
Line 64: should be knock-out not knock-down.
Line 71: In this referenced paper, mosquito population crash was not simply caused by dsx knockout but also combined with a gene drive system, which rapidly spread the disrupted dsx allele into the population. It'll be better if the authors can specify this point here.
Line 81: This sentence should be revised as "The sex-specifically expressed genes..."
Line 83: again, should be knockout.
Line 87-88: This sentence needs to be revised to “In B. mori, the functional importance of dsx was verified and its potential application in SIT was proposed”.
Line 89-91: “In A. gambiae, researchers found that introducing a sex distorter female specific lethal Fle gene and the gene drive of dsx could cause progressive decreased ratio of female and eventually collapse the population..” will be better to be revised to “Novel population genetic control methods, gene drives, targeting female-specific lethal genes have been reported in A. gambiae, such as a sex distorter targeting Fle and a gene drive targeting dsx, which caused progressive decreased ratio of female and eventually collapsed population”.
Line 97: “..specific genes of dsx..” should be “specific transcript of dsx”.
Line 99: “..are observed..” should be “were observed”.
Line 113-115: The identification of a homolog in a different species should be done with the reciprocal protein-protein blast. If this is what the authors did, this sentence needs to be corrected as “The dsx of S. frugiperda was identified using the blast with amino acid sequence of D. melanogaster (GenBank accession number NP_001287220.1) and B. mori (GenBank Accession number NP_001036871.1) against S. frugiperda database (please specify which database the authors used)”. Additionally, the hits in S. frugiperda should also be blasted against D. melanogaster and B. mori database to confirm the homolog.
Line 118: the authors should specify how they designed these gRNAs. For example, which online tool or what design strategy they applied.
Line 124: “..with protospacer adjacent motif (PAM) sequence localized upstream”. The PAM site is not needed for designing in vitro transcription template, but gRNA scaffold and a poly T terminator is required.
Table 1: It is confusing to list the aims (e.g. identification of somatic mutations) in the same column with primers. It will be better to list them in a separate column. Also, I suggest moving Table 1 to supplementary data instead of in the main manuscript, since it is not very important for explaining results.
Line 132: there should always be a space between the unit and the number (e.g. 300 ng/μL, instead of 300ng/μL). Please also revise the rest of the manuscript.
Line 153: why did the authors use two-day old dsx mutants instead of freshly emerged adults? Were these adults reared separately before sexing and mating? What I am concerned is, if adults mate with their siblings (if they were reared together) within the first two days, the fecundity/fertility analysis might be affected.
Line 165: authors should specify which primer pairs were used for RT-qPCR of different genes. Was the same primer pair used for RT-qPCR of dsxC, dsxM and dsxF? Was these primers in the common region of both male and female dsx?
How many “SfdsxC male and female, SfdsxF female, and SfdsxM male” mutants were used for RNA extraction? Were their genetic mutations confirmed by PCR?Or were they simply individuals injected with corresponding gRNAs (which showed mutant phenotype but was not genotyped)?
Line 192: again, blast should be done with amino acid sequence.
Line 200: “post birth” should be corrected with “post oviposition”.
Line 206: “puparia” should be “pupa shell”.
Line 211: should be “...was significantly increased..”.
Figure 1: Do Figure 1B and 1C both show mutant sequences detected with pJET cloning? If so, they should be merged together. Actually, C alone would be enough to show the mutations caused by Cas9 editing.
Besides, the Cas9 cleavage site is 3-4 nucleotides upstream of the PAM, which is not what red arrows pointed at in the figure. I suggest deleting those arrows.
The relative expression level should not have a unit, thus please delete %.
Normally, the significant difference is indicated with P<0.05. Please use the same threshold of significant differences with that in Figure 5.
“one-factor ANOVA” should be “one-way ANOVA”.
Line 246: “fissure” should be “figure”.
Figure 2&3: The exact sex of these mutants should be confirmed by genotyping (e.g. PCR amplification of W-chromosome specific genome sequence). It will make the conclusion (abnormal genitalia was caused by disrupting sex-specific transcript) more solid.
Figure 4: arrows are white not red.
Figure 5: experimental group Sfdsxc female x WT male is missing. The corresponding data should be added. It is not clearly marked in the figure that which groups showing significant differences.
Line 324: Considering about the potential impact of dsx deficiency to mating and reproductive behaviour, the expression difference of pheromone receptors among mutants and wildtypes could be interesting to readers. It would better to show the result in the main context instead of supplementary data.
Line 357: please delete “(Noctuidae)”.
Line 373: should be “knockout” not “knockdown”.
